# Query Variability and Experimental Consistency:
# A Concerning Case Study

## ABSTRACT

In offline experimentation, the effectiveness of a search engine is evaluated using a document collection, a set of queries against that collection, a set of relevance judgments connecting the documents and the queries, and an effectiveness metric. This measurement pipeline is used as a surrogate for user satisfaction – the extent to which the system provides useful information to the users issuing the queries. But queries are responses to information needs, or topics, and there can be a wide variety of ways in which any given information need can be expressed as a query. That one-to-many relationship suggests that, in an IR experiment, use of any single query to represent a topic may be insufficient. In this case study, we demonstrate that this practice is indeed a weakness. We show that the TREC 2013 and 2014 Web track queries, which are regarded as being indicative of specific information needs, are not representative of crowd-generated queries for the same underlying needs, and can give rise to inconsistent system relativities when compared to user-generated queries. From this instance we must thus note a clear concern: that current test collection design strategies can lead to effectiveness results that are at odds with those experienced by typical non-expert users.

## CCS CONCEPTS

• **Information systems** → **Evaluation of retrieval results**; **Test collections**; **Relevance assessment**; **Retrieval effectiveness**.

## KEYWORDS

Evaluation; significance testing

**ACM Reference Format:**
Anonymous Author(s). 2024. Query Variability and Experimental Consistency: A Concerning Case Study. In *Proceedings of the International ACM SIGIR Conference on the Theory of Information Retrieval, July 2024, Washington, USA.* ACM, New York, NY, USA, 7 pages. https://doi.org/XXXXXXX.XXXXXXX

## 1 INTRODUCTION AND BACKGROUND

In offline IR evaluation, the quality of the results retrieved by a search engine is assessed using a collection of documents, queries, their corresponding relevance judgments, and an effectiveness measure; see Sanderson [20] for more information. While all aspects

of this framework are necessary, the queries play an especially important role. If they are to be a realistic test they should represent the way in which one or more users would formulate the retrieval task in response to a given information need. Since this is a human process, there is no single query that can be regarded as being definitive for any given information need; for this reason, there has been a range of work that has examined user query variability [2–4, 7, 17, 21, 25], the effect that query variations have on pooling costs [16], and ways in which query variations might be exploited to boost retrieval performance [5]. While a range of techniques such as query suggestion, substitution, and expansion are used to refine and, to a certain extent, homogenize the initial query in order to produce better retrieval results [8, 14], there is nevertheless the potential for considerable variability in the effectiveness experienced by different users.

The significance and impact of query variations is at odds with the design of many of the ⟨documents, queries, judgments⟩ corpora that have been widely used by the IR community. For example, many of the TREC Ad Hoc and Web Tracks over the last three decades of measurement have relied on the use of a single query associated with each information need, with the implicit assumption that this single query is canonical. The relevance judgments have been guided by a narrative that provides additional and nuanced information about what is required in order for a document to be an answer, but the TREC-supplied queries – normally one per topic, and sometimes denoted as being the topic "title" – are usually the ones used to compare system effectiveness.

That is, in many experimental contexts there is no inclusion of query variations as a factor that might affect experimental outcomes. Furthermore, not only is the use of only a single query of concern, but in some corpora the query used is created as a topic descriptor – a quite different function to, say, choosing a phrase because it is believed to be a "typical" query for that topic.

It is those concerns that prompted the investigation reported in this case study. First we consider a very simple question: *if one user perceives some retrieval system (System A, say) to be better than another (System B) over a set of queries, will a different user with the same set of information needs, but expressed via different queries, also perceive System A as being better?* That is, we examine the extent to which system comparisons are agnostic to the query formulations used to measure them.

Reassuringly, using the TREC-supplied queries for the 2013 and 2014 Web track (for which a large pool of query variants was created for another study) we find that the systems that perform well using one query variant for each topic are also likely to perform well if presented with a different set of user-generated queries. That is, systems that appear to be "good" to one user do indeed tend to also appear "good" to other users.

We then consider a related question: *is the same consistency observed if one of the "users" is in fact the set of TREC-supplied*

*queries?* The answer is concerning, as our results show that the TREC-provided queries are exceptions to the general pattern. Relationships between systems that are established via the TREC-provided queries are often *not* supported by user-generated queries that address the same information need. That is, this second experiment demonstrates that the TREC queries, upon which so much reliance is placed in terms of IR experimental methodology, may *not* be reliable predictors of the system–versus–system relationships that might be experienced by users formulating their own queries.

To summarize: there has been an implicit assumption in the use of many TREC test corpora that the TREC-provided query for each topic is representative. Using the one corpus for which suitable user-generated queries and relevance judgments are available, we describe an approach in which we are able to test this widely held evaluation assumption, and show that the TREC-provided queries have different statistical properties to worker-generated queries. Our case study is limited to that single corpus; nevertheless it provides a counter-example to that implicit assumption, and hence raises clear concerns that require careful consideration.

## 2 EXPERIMENTS

We now describe the experiments that were carried out and present their outcomes. To address our questions we require a standard test collection of topics, documents, and judgments, plus in addition require for each topic a set of query variants. Such variants are available only for the TREC 2013 and 2014 Web tracks, described shortly. We also require selection of retrieval systems, as we wish to examine relative performance between systems as a basis for determine whether relative performance is preserved when different query variants are used.

**Queries and Systems.** As noted, we make use of the TREC 2013 and 2014 Web track resources. The queries used in these tracks were from commercial search engines, and selected as being representative of typical search tasks. For each of TREC 2013 and 2014, a set of 50 such queries was collated, taking a mix of broad and specific information needs. To prevent ambiguity, we refer to these as being the "*seed queries*" associated with the corpus for these two tracks. Some of the seed queries are structured in a multiple-subtopic paradigm, while others are focused on a single subtopic [10, 11].[1]

We use the TREC 2013 and 2014 corpus because subsequent work created query variants, known as the UQV100 queries, together with relevance judgments derived from document pools formed using those sets of query variants [2]. That process commenced by creating a set of 100 information need statements (referred to as *backstories*) developed from the 100 TREC-supplied seed queries for the 2013 and 2014 Web tracks, selecting (where there were multiple options available) one identified subtopic. The backstories can thus be thought of as inferred topic statements, akin to the topic descriptions that were used in earlier TREC rounds. For consistency with previous work, we continue to refer to each of these inferred information needs as a topic.

Those backstories were then presented to crowd-workers, who were asked what query they would use in response to each given

information need, a process that led to a total of 10,835 query variants being collected, averaging (after data validation and filtering) 108 queries per topic, and 57.6 distinct query variants per topic [2]. High levels of diversity in user-generated queries in response to backstories has been a key finding of work in this area [2, 17].

We then checked the 100 sets of user-generated UQV100 queries, to see if the corresponding TREC-provided seed queries had been suggested as a query by the crowd workers. There were 77 queries for which that had happened. Because we were interested in comparing the seed queries and user-generated queries, we selected that subset of 77 for use in our experiments. That is, each of the 77 topics employed in the new experiments described shortly has a user-generated query set that includes the corresponding TREC seed query, and hence has relevance judgments for which the TREC seed query was a "first class contributor". That filtering process led to a total of 4,218 distinct queries, or 54.8 per topic.

Those queries were then passed to a suite of fifteen different retrieval systems, and retrieval runs prepared. The fifteen systems used three different ranking models: BM25 [19] (a probabilistic retrieval model based on bag-of-words); QLD [23] (query likelihood with a Dirichlet smoothed Language model); and SPL [9] (an information theoretic model). Other variants of these three core systems were created via an RM3 query expansion model [1] and via axiomatic reranking algorithms [12], using different parameter settings. We used the Anserini toolkit [22] to formulate these search engines with their different extensions and parameter settings.[2]

We make use of several effectiveness metrics and hence a range of different corresponding user models, to cover the spectrum of possible evaluation scenarios. The five metrics employed are average precision (AP), a top-weighted recall-sensitive mechanism [6]; normalized discounted cumulative gain (NDCG) [13], which is also recall-sensitive but less heavily top-weighted; precision at depth 10 (Prec@10), modeling users who always look at exactly ten results; rank-biased precision (RBP) with a parameter $\phi = 0.85$, simulating users who on average view the top approximately seven answers in each results listing [15] but may also look at fewer or more; and reciprocal rank (RR), modeling users that search until they find a first relevant document. The last three have associated user models that do not require the user to be aware of the total volume of possible answer documents in the collection [15].

**Experiment 0: Validation of Systems and Judgments.** The relevance judgments for the UQV100 collection were generated by pooling the runs for each topic (over an average of 57.6 distinct queries per topic), with all runs generated at first by an Indri/BM25 system [2], and then later augmented by runs from four other retrieval systems.[3] Multiple rounds of judging were undertaken: first, uniform pooling was applied to depth 10 on each run of the original Indri/BM25 system, yielding 21,895 judgments (covering all 100 topics); then a further 5,501 documents were judged based on a "weighted coverage" basis [2]; and finally further judgments were undertaken when the query runs from another four research retrieval systems were included. When restricted to the 77 topics selected for our experiments, this total set of 55,587 judgments was

---

[1]As well, relevance judgments relative to the seed queries and their possible subtopics were created by NIST assessors using a six-point scale [10, 11], but we did not use those judgments in these experiments.

[2]https://github.com/castorini/anserini
[3]See http://dx.doi.org/10.4225/49/5726E597B8376, file README.txt.

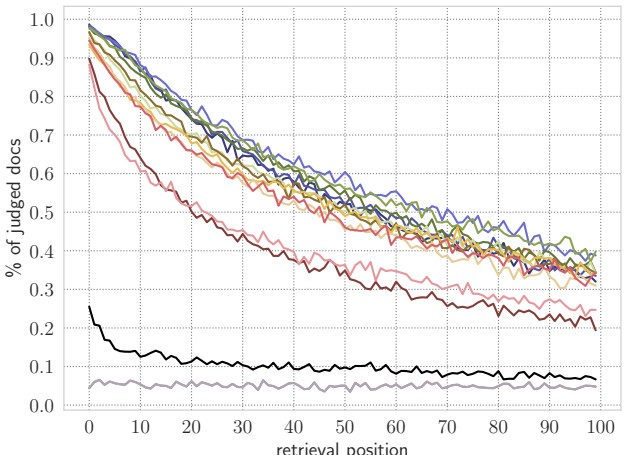

**Figure 1:** Fraction of judged documents at each position in runs, averaged over 4,218 queries associated with 77 topics. Each line represents one of the fifteen systems constructed.

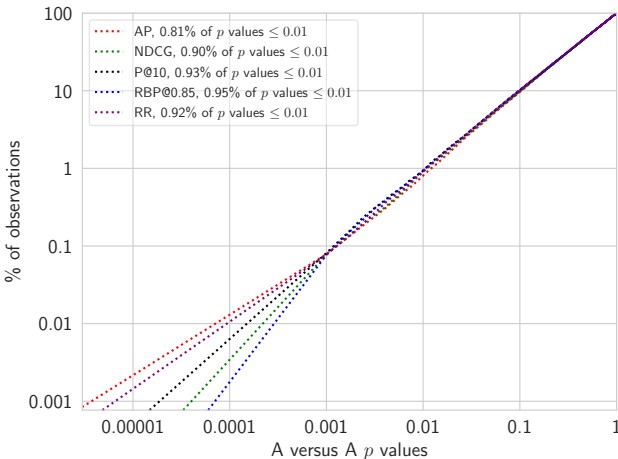

**Figure 2:** Comparing a system with itself (Experiment 1). A $p$ value is generated by comparing the scores observed by user $\alpha$ and user $\beta$ when they both use the same system to process different queries that address the same original information need. The annotations in the legend show that for all five effectiveness metrics the number of false positives is below the computed $p$ value when summed for all user pairs for which $p \leq 0.01$.

reduced to 39,478 judgments (71.0% of the original set), of which 8,140 (20.6%) were relevant at grade one or above.

To validate the judgments, and to verify that they were suited to the fifteen systems we employed, we first computed, for each system, the fraction of judged documents that were surfaced by the query variations at each position in the run. Figure 1 shows the results. Thirteen of the systems exhibit what we would regard as "normal" behavior, with high average fractions of documents judged at early positions in the corresponding runs, tapering downward as the depth in the run increases.

But two of the systems were notably anomalous. Those two systems surfaced quite different document sets, with very low fractions judged, and hence with correspondingly low accuracy in any computed effectiveness scores. As a result of this step we removed those two systems,[4] and proceeded to our main experimentation using the remaining thirteen systems, satisfied that the UQV100 relevance judgments provide a reasonable fit.

**Methodology.** To simulate a *pair of random users* we sample the set of 4,218 query variations on a per topic basis, selecting two different variants for each topic, and assigning one to user $\alpha$ and the other to user $\beta$. This gives us a sequence of 77 queries for user $\alpha$, and a set of 77 disjoint queries for user $\beta$, but with both $\alpha$ and $\beta$ able to be regarded as seeking answers to the same set of 77 information needs. That is, user $\alpha$ and user $\beta$ can be considered as a paired experiment for statistical testing purposes, to determine if $\alpha$ receives better quality responses from a system than does user $\beta$.

The sampling and statistical testing process can then be repeated many times, to develop an overall pattern of behavior, in a manner akin to the bootstrap test. In the experiments reported next a total of 10,000 repetitions were carried out, each involving a user $\alpha$

---

[4] Note that we regard interrogation of experimental outcomes in this way as a critical and valid step prior to making inferences about results. In particular, there is no basis for assuming in experiments of this form that unjudged documents are irrelevant, and the presence of systems where there is high uncertainty in measured results is a confound. On the other hand, removal of any particular system from the results does not introduce bias.

searching using a set of 77 queries, paired with a user $\beta$ searching for the same information but via a different set of 77 queries.

**Experiment 1: A versus A.** In this experiment we suppose that user $\alpha$ and user $\beta$ both make use of the same retrieval system to process their queries. We employ each of the 13 retrieval systems, and 10,000 drawings to form pairs of users; that is, we in effect carry out 130,000 experiments in which two different users are assumed to use the same system to search for the same information need.

Each of those trials generates paired vectors of 77 metric scores for each of the five effectiveness metrics, and hence can be further processed by a statistical test to determine a set of five $p$ values. Having always compared a system against itself, we expect to see very few findings of "there is a significant difference between the experience of user $\alpha$ and the experience of user $\beta$" (when "experience" is assessed via the corresponding metric score) that is, very few small $p$ values. Figure 2 shows the results (using a Student $t$-test) and confirms that we have generated the expected pattern of results. Each of the five lines represents cumulative fractions for one of the effectiveness metrics, and while they differ a little at the left-hand low-frequency section of the plot, they are closely aligned through the region of primary interest between $p \approx 0.001$ and $p \approx 0.05$. For example, regardless of metric, approximately 1% of the experiments resulted in a $p$ value less than 0.01.

This outcome validates the methodology we have developed, and demonstrates that the five metrics all display the expected behavior.

**Experiment 2: A versus B.** A frequent goal in offline IR evaluation is to compare two systems, evaluating an incumbent regarded as being a *champion* against a newer *challenger*. With two systems in play, referred to here as being "*A*" and "*B*", and two users $\alpha$ and $\beta$ with the same set of information needs but different queries, as

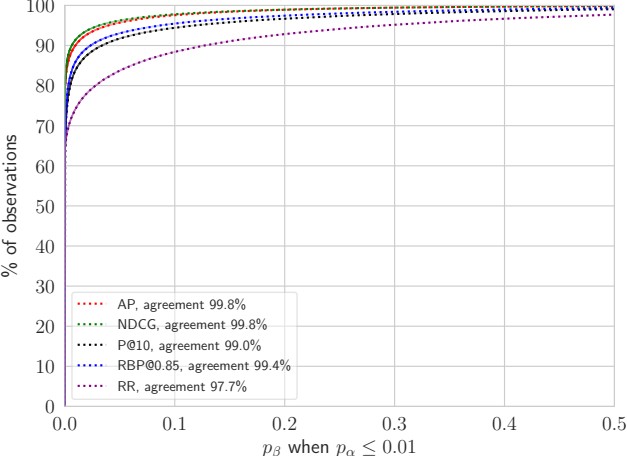

(a) Cumulative distributions for $p_\beta$, given that $p_\alpha \leq 0.01$

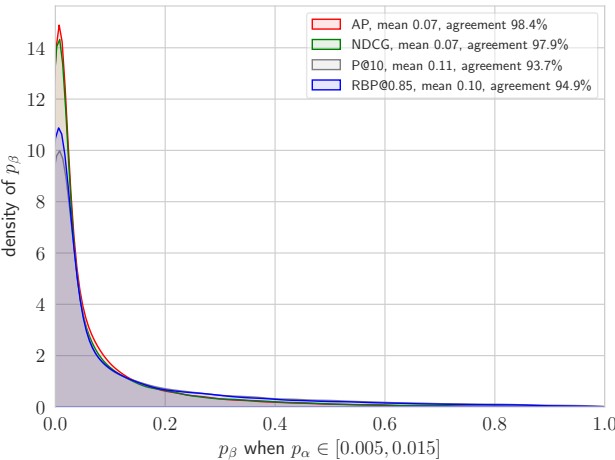

(b) Density of $p_\beta$, given that $0.005 \leq p_\alpha \leq 0.015$

**Figure 3:** Comparing systems (Experiment 2), $A$ versus $B$ outcomes over 10,000 trials each consisting of 77 randomly selected queries. In plot (b) when $p_\beta < 0.5$, System $A$ is superior to System $B$; when $p_\beta > 0.5$, System $B$ is superior to System $A$, with $1 - p_\beta$ plotted instead, thereby forming a single continuous scale.

already described, we can ask the contingent question "*if user $\alpha$ observes better performance from System A than they do from System B, will user $\beta$ observe the same relationship?*"

In the first part of the exploration we restrict our attention to system pairs $A$ and $B$ for which user $\alpha$ detects a strongly statistically significant outcome over their set of 77 queries in favor of System $A$, filtering the set of all possible $(A, B, \alpha, \beta)$ tuples to the subset for which user $\alpha$ calculates a Student $t$-test $p_\alpha$ value that is $\leq 0.01$. Applying that filtering process on a per-metric basis reduced the 780,000 $(A, B, \alpha, \beta)$ (starting with 10,000 query sets $\alpha$, and with 78 system pairs possible from 13 systems) combinations to between 191,115 (for RR) and 482,367 (for NDCG) combinations.

Working only with those subsets, we then ask what $p_\beta$ value was observed by user $\beta$ when comparing the pair of systems. The

results are depicted in Figure 3(a), with a cumulative distribution of $p_\beta$ plotted for each of the five effectiveness metrics, and with the detailed values in the legend recording the fraction of $p_\beta$ values less than 0.5. (In these plots values of $p_\beta > 0.5$ indicate that user $\beta$ measured System $A$ as being *inferior* to System $B$.) That is, those annotated "agreement rates" reflect the total fraction of tuples for which user $\beta$ also observes System $A$ to be superior to System $B$.

With $p_\alpha \leq 0.01$, we expect that final agreement fraction to be $\geq 99\%$, and that is indeed what occurs for four of the five metrics, as noted in the graph's legend box. On the other hand, RR only achieves a 97.7% "predictive score" in this experiment, an outcome that is not surprising given that RR is not suited to the Student $t$ test. (That is, because the paired score differences that RR generates are unlikely to be normally distributed; thereby supplying a timely reminder in regard to statistical tests only being valid if their preconditions are satisfied).

As a further observation, note that the upper pane in Figure 3 also shows that more than 80% of the $p_\beta$ values are not just less than 0.5, but are also smaller than 0.01, meaning that 80% of the time user $\beta$ would also find that System $A$ was significantly better than System $B$ at the $p_\beta \leq 0.01$ level.

In the second part of the experiment we restrict $(A, B, \alpha, \beta)$ in a slightly different way, taking those combinations that yield $0.005 \leq p_\alpha \leq 0.015$ for each metric, that is, a band of broadly comparable significance outcomes centered around 0.01. This alternative filtering process reduces the 780,000 $(A, B, \alpha, \beta)$ combinations to between 39,263 (for NDCG) and 43,984 (for AP) combinations. The results are plotted for four of the metrics in Figure 3(b) as an inferred density distribution associated with the corresponding $p_\beta$ values, calculated using the Kernel Density Estimation function in the Python seaborn package.

We again see the expected behavior – for each of the metrics the $p_\beta$ density peaks in the vicinity of 0.01, indicating that users $\alpha$ and $\beta$ observe broadly similar outcomes when comparing System $A$ and System $B$, even though they distilled the information need into different queries. Moreover, while the $p_\beta$ values observed by user $\beta$ have means in the range 0.07 to 0.11 (noted in the legend box), and are almost ten times larger than the corresponding $p_\alpha \approx 0.01$ values, this is not of concern. Statistical significance on the part of user $\alpha$ does not imply that user $\beta$ should see the same level of significance, only that $\beta$ is likely to observe that System $A$ is superior when the $A$'s mean is compared to $B$'s mean. The levels of agreement also drop in this part of Experiment 2 – in the case of Prec@10 and RBP quite notably so – but this is also not problematic in any way. In removing the tuples $(A, B, \alpha, \beta)$ in which $p_\alpha < 0.005$ the expectations in regard to the fraction of times agreement should be observed by user $\beta$ are weakened, and weakened by, as it turns out, different amounts across the suite of effectiveness metrics.

The results in Figure 3 answer the main question posed in Section 1: *if one user perceives some retrieval system (System A, say) to be better than another (System B) over a set of queries, a different user with the same set of information needs, but expressed via different queries, is also likely to perceive System A as being better.*

**Experiment 3: Query Variations versus Seed Queries.** Our next experiment compares TREC seed queries against the other query variations in the UQV100 test collection. In this experiment

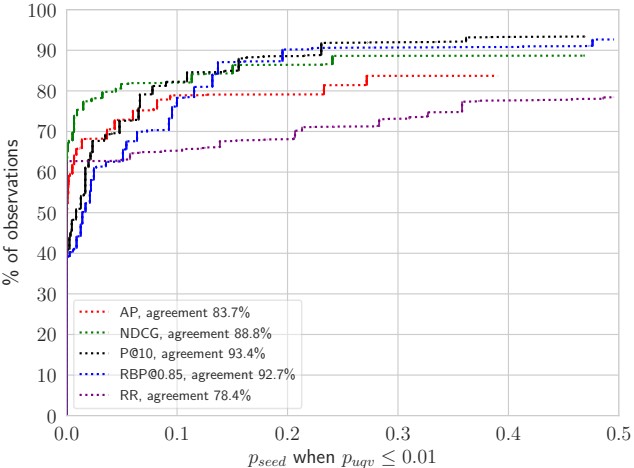

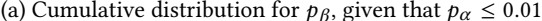

(a) Cumulative distribution for $p_\beta$, given that $p_\alpha \leq 0.01$.

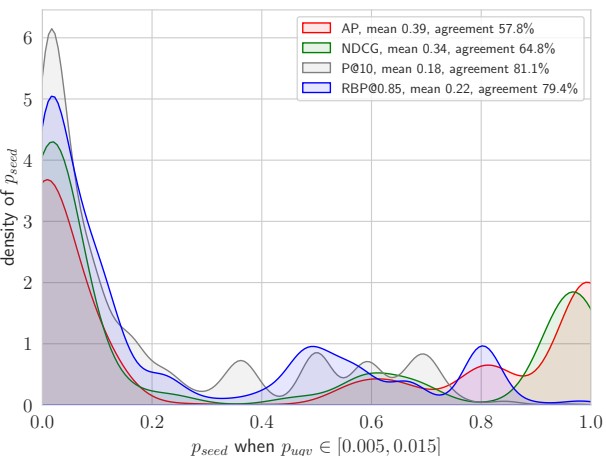

(b) Density of $p_\beta$, given that $0.005 \leq p_\alpha \leq 0.015$

**Figure 4:** Seed versus UQV queries (Experiment 3). User $\alpha$ employs a randomly selected worker-generated query for each topic; user $\beta$ always uses the TREC-supplied seed query for each topic. Other details are as for Figure 3.

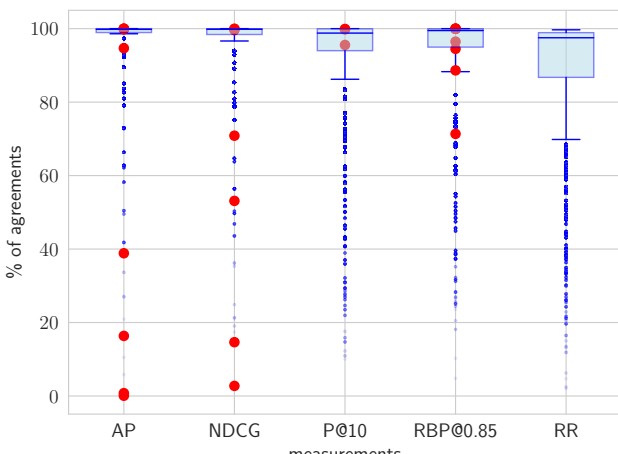

**Figure 5:** Distribution of agreement rates between sets of random user queries taken as $\alpha$ users (all blue dots, and the blue box-whisker elements) and 10,000 other query sets; and between TREC seed queries (red dots) as the $\alpha$ set and the same set of 10,000 $\beta$ query sets. In all cases the $\alpha$ set led to significance between System $A$ and System $B$ with $0.005 \leq p_\alpha \leq 0.015$.

the 77 queries for user $\alpha$ are a random sample from the UQV100 query set, including the TREC seed query for each topic, and we apply the same two filtering options on tuples $(A, B, \alpha, \beta)$: inclusion when $p_\alpha \leq 0.01$; and inclusion when $0.005 \leq p_\alpha \leq 0.015$. What makes this experiment different is that user $\beta$ is assumed to *always* employ the TREC-provided seed query, recalling that the 77 topics were selected because the seed query was amongst the options proposed by the crowd workers.

Figure 4 presents the same two views as in Figure 3, but with clear differences visible. In the upper pane in Figure 4, the agreement levels are lower than those shown in Figure 3(a). For example, even putting the RR curve to one side because of the mismatch between it and the Student $t$-test, if user $\alpha$ employs their metric of choice and detects a difference between System $A$ and System $B$ at the 0.01

level, that "significant" relationship would not be observed by user $\beta$ between around 5% and around 15% of the time.

The lower pane in Figure 4 confirms this lack of predictivity. In this plot the value 0.01 on the horizontal axis indicates that user $\beta$ finds System $A$ to be significantly better than System $B$; similarly, the point 0.99 indicates that user $\beta$ observes the *reverse*, that System $A$ is significantly *worse* than System $B$ at the 0.01 level. That is, in this graph (and also in Figure 3(b)) we have created a single "blended $p$ value" scale that spans the range from 0 to 1, and similarly spans the spectrum from System $A$ being better through to System $B$ being better.

The regions of moderate density for AP and NDCG at the right hand end of this lower plot are what are most startling. They indicate that it is not at all uncommon for user $\alpha$ to claim that System $A$ is significantly better than System $B$, but for user $\beta$, who has used the TREC seed queries, to simultaneously believe – and just as strongly – that they have assembled evidence that System $B$ is better than System $A$.

**Experiment 4: Patterns of Agreement.** Figure 5 consolidates Figures 3(b) and 4(b) into a single presentation. To make the blue box-whisker elements in this graph, each of the $\approx$ 40,000 $(A, B, \alpha)$ combinations (of 780,000 as a starting point) for which $0.005 \leq p_\alpha \leq 0.015$ was regarded as a "reference user", and each one of those was compared to a further set of 10,000 $\beta$ selections. Each point plotted in each box-whisker element is then the fraction of those 10,000 trials for which the set of $\beta$ users observed the same ordering relationship between Systems $A$ and $B$ as did user $\alpha$. There are thus $\approx$ 40,000 such points (the exact number varying according to the metric, as noted above) plotted in each of the five box-whisker elements. That is, the box-whisker elements show the distribution of the agreement values that are condensed into the overall averages given in the legend box of Figure 3(b). The

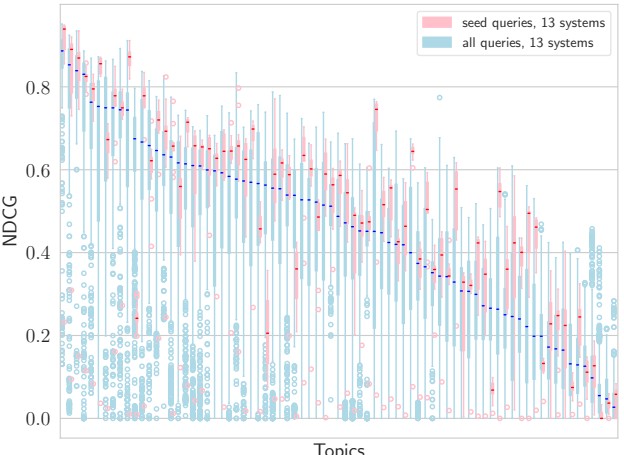

**Figure 6:** Absolute NDCG score distributions using all query variations per topic and 13 systems (blue), for each of 77 topics. The second distribution (pink) for each topic reflects the use of the seed queries across the 13 systems. Each pair of adjacent blue and pink bars represents one of the topics, ordered by the median (across systems) blue point.

distributions are reasonably tightly centered on the median values, shown as solid lines in the boxes, but in each case there is also a long descending tail of outliers. Each outlier represents a reference user who observed – with strong statistical significance – a System *A* versus system *B* outcome that varied markedly from the aggregate view of 10,000 other users who queried using other variations.

The red dots in Figure 5 then reflect the same measurement, but with $\alpha$ always the set of seed queries, and with *A* and *B* a system pair that are differentiated by the seed queries with $0.005 \leq p_\alpha \leq 0.015$. (For example, of 78 possible system pairs, 7 are retained when the metric is NDCG; recall that we are examining only a relatively narrow band of $p_\alpha$ values in this experiment. There were no system pairs fitting this criteria for RR). An agreement rate is again computed from 10,000 randomly generated $\beta$ query sets, but now we are asking, "*if two systems are significantly different according to the TREC seed queries, what fraction of user-generated query sets will obtain the same system relativity?*" The difference between the random pairings and the seed query pairings is now stark. The TREC seed queries, even though they do sometimes arise from the crowd worker elicitation process (in particular, in all of the 77 topics used in these experiments), form a query suite that when taken as a collective whole is distinctively different from the worker-supplied queries.

**Experiment 5: Absolute Performance of Seed Queries.** It is also interesting to consider whether the seed queries give rise to better retrieval effectiveness than do the user-generated queries. Figure 6 plots NDCG scores, and shows that in general they do. The spread of per-topic NDCG scores across the 13 systems and total of 4,218 queries in the blue bars is very diffuse, and some of the user query variations lead to very poor performance. Only a relatively small fraction of that broad query set out-perform the seed queries, but nor are the seed queries the best for most of the topics.

## 3 CONCLUSION AND FUTURE WORK

We have used the UQV100 data to explore whether TREC-supplied seed queries are representative of worker-generated queries for the same topics, noting that the seed queries – that get used in many IR experiments – are but a single way in which the underlying information needs might be represented. The UQV queries pertain to the TREC 2013 and 2014 Web track, and thus provide a case study in which the representativeness of seed queries can be tested. This is currently the only combination of resources with which such an experiment can be carried out.

Focusing on a subset of the UQV100 topics for which the seed query was also provided by one or more crowd workers, and using a total of thirteen different retrieval systems, we have measured the predictivity of statistical tests when assessing experiments making use of one query version per topic. When query variations are compared against each other via random sampling, all is well – if a paired statistical test reports for one user that two retrieval systems are different, a second user making their own selection from the queries is highly likely to identify the same relativity between the two systems.

But when one of those two users always employs the corresponding TREC seed query, and the other issues a worker-generated query other than the seed query, the situation is more complex. Taken as a specific subset, the seed queries often give rise to different outcomes to the UQV100 query set, and a user who bases their evaluation on only the TREC-provided seed queries will, more often than can be accounted for by random fluctuations, observe reversed system relativities relative to a user who samples from the pool of query variations. Indeed, a TREC 2013 and 2014 "seed-only" querier may well find that they have evidence in support of System *A* being statistically superior to System *B*, while at the same time a non-seed querier might have equally compelling evidence in favor of System *B*. That is, use of the TREC 2013 and 2014 seed queries alone in a challenger versus champion experiment could lead to an outcome that is at odds with the relativity observed by typical users of those same two systems.

We further note that these results are based only on the 77 topics for which one or more of the UQV crowd workers generated the TREC seed query in response to the topic's backstory. The comparison might be even more divergent for the other 23 topics, for which the crowd workers have implicitly indicated that the seed query is *not* a popular choice.

This negative finding provides further support to the encouragement already articulated by Bailey et al. [2, 3] for researchers and practitioners alike to make use of multiple queries per topic; and to ensure that the judgments being used in experiments are equitably suited to the evaluation of all of the experimental systems. While we again acknowledge that we have explored only a single collection, this one test already provides a counter-example to the assumption that systems can be reliably evaluated on TREC-specified seed queries alone. If and when other collections have similar query variations and suitable relevance judgments created, it will become possible to consider the prevalence of the problem we have documented here. But one such instance – the case study of this paper – is sufficient to establish that the problem can and does arise.

In terms of future work, we note that what we have done here can be thought of as "query bootstrapping", while holding the set of topics constant. From that point of view it complements the previous work by Rashidi et al. [18] that explores "judgment bootstrapping", and by Zobel and Rashidi [24] that explores "corpus bootstrapping". That observation then opens the possibility of combined bootstrapping modes, in which more that one of these elements is varied, so as to further investigate experimental predictivity.

**Acknowledgment.** Will be provided in the final version.

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
