# OpenReview forum: "Query Variability and Experimental Consistency: A Concerning Case Study"
_ACM.org/SIGIR/ICTIR/2024/Conference — ICTIR 2024_

### Official Review · Reviewer_EuAr · 2024-05-06

**Rating:** 2
**Confidence:** 5

**Objective Part Of Review:**

The abstract, methods, and results are clearly stated.  The central claim of the paper, "current test collection design strategies can lead to effectiveness results that are at odds with those experienced by typical non-expert users", is supported but the results show that the answer is somewhat more nuanced.  I did not find concepts or notations that were used prior to being defined, but that may be an artifact of my familiarity with the data and methods.  The abstract is extremely clear and definitely provides an executive summary of the work.

The abstract and the text state that the TREC 2013 and 2014 Web Track topic "query" fields are not representative of the [UQV] crowd-generated queries, and strongly implies that the queries are not representative of queries from actual users.  According to the TREC 2013 and 2014 Web Track Overviews (see https://trec.nist.gov/pubs/trec22/papers/WEB.OVERVIEW.pdf and https://trec.nist.gov/pubs/trec23/papers/overview-web.pdf) state that the "query" fields are actual user queries from a query log from a major search engine company.  Perhaps the crux here is the word "representative" which is not defined statistically.

The paper also implies that the "title" field of a TREC topic is always a query.  This is not the case, and in these two tracks there is actually no "title" field, but rather a "query" field.  As Voorhees and Harman state in "TREC: Experiment and Evaluation in Information Retrieval" (2005; p28) the topics are intentionally -not- queries ("requests" as Harman puts it there) for several reasons, one of which was "a desire to allow a wide range of query construction methods".  In those days I suspect they were thinking of free-text vs structured queries or Boolean queries, but clearly this paper is showing precisely WHY there are not queries in the TREC topics.

The authors of the UQV collection collected relevance judgments on pooled baseline runs, so there is an element of assessor variation that should be noted.  Voorhees (SIGIR 1998) shows that there is low overlap (~40% using Jaccard overlap) but that the overall ranking does not change much.  In UQV the topics are accompanied by a "backstory"... can the backstory change the nature of relevance enough to induce differences from TREC's assesses relevance?  Could the result of the paper on the TREC "query" vs the UQV queries be attributed to assessor variation?

If we take the UQV queries as a true random sample of queries, we should be able to use these results to express a confidence interval on the retrieval score encompassing the variance due to the query.  That metric would be useful in general if it can be estimated for other collections as well, since (as this paper claims) systems in each others intervals should be thought of as indistinguishable given query variation.

The plots in figure 3 and 4 show the cumulative distributions for p_\beta (the p value of a Student's t-test for two runs in the beta sample) given a fixed p_\alpha.  Surely there are many fewer significant results when the p values are close to zero.  Is the size of the sample there driving the high variance?

**Subjective Part Of Review:**

This paper follows in the steps of a series of very interesting papers inspired by the UQV effort.  The authors of this paper contribute to the conclusion that query variation exists, can be large, and should be measured.  I think the risk of using the results of TREC title-only runs as representative is small, but I have not measured that across all TREC collections.

The web track topics are all built from search engine query logs.  As such, the queries there come from a different period of time than the UQV crowdsourced queries.  I wonder if there is also a variation in time here, where query formulation practices have changed.  Since users are not usually aware of their query formulation process (unless they are an IR or IS researcher ;-) or how the search engine encourages and optimizes those processes, this is maybe unknowable.  I think if LLMs usher in a new era of very long queries then the results on short TREC queries will be much less usable.

I would love to be able to generate UQV-like query sets given a topic statement (and maybe a backstory).  I have tried this with LLMs and don't think this is a good idea at the present time, but perhaps in the next generation of models.  Part of this is this notion of "representativeness" which is ill-defined.  (Most of it is that current LLMs don't like to give lots of different answers.). If systems do query generation and fusion under the hood, are current single-"query" collections better off, or worse off?  If the collections also include large sets of plausible queries for each topic, will that be enough to capture query variance?  (esp when the systems, as above, are themselves generating query variants)

---

### Official Review · Reviewer_xFBr · 2024-05-13

**Rating:** -1
**Confidence:** 3

**Objective Part Of Review:**

This paper, submitted to ICTIR 2024, is about the influence of different query formulations on the validity of offline IR experiments (in this case, mainly the TREC 2013/14 Web Track). The authors compare the difference between retrieval experiment outcomes based on the different query formulations in the UQV100 dataset that complements the previously mentioned Web Track. The outcomes show that while high compatibility of the outcomes is given, when different uqv are tested against each other, these similarities in the outcomes fade away when these query variants are compared against the "official" TREC queries.

The results of the case study's experiments are clearly stated in the introduction, and the overall issue is also clearly stated.

I am not convinced of the general assumption of the study "that systems can be reliably evaluated on TREC-specific seed queries alone." Yes, there are test collections that only have "queries" instead of "topics". But as long as you have a topic it's best-practice to make use of description or narrative to form a better query. Using title-only queries generally does not lead to the best results in classic ad-hoc experiments. And even if there are title-only queries, these get expanded or extended in many ways. A setting that Buckley and Walz already described and worked on in the TREC-8 Query Track.

To make their initial motivation stronger, the author might include some numbers on typical test collections and their use of seed queries, etc.

There are some other unclear parts in the papers, that I would like to list:
- What are the 15 different system variants? How are they combined using the three ranking models?
- Why are only sparse representation models considered? Would dense representations (using contextualized language models) perform differently?
- Why is there no bias introduced when discarding two systems (we don't know which systems those were - would have been interesting to see what might have caused this different behaviour).
- The text includes many numbers that are hard to get/align like 13/15 systems combinations, 5 metric, 78 pairings based on 4218 query variations... and many other numbers that make it confusing to follow. A table might help to provide a better overview.
- It's hard to distinguish text parts that describe other's work (like to construction of UQV100 or section "Experiment 0") and the author's contribution. I had to re-read many paragraphs because of that. A own section on "data sets" or "methods" might clarify this better.

**Subjective Part Of Review:**

I totally miss the "theoretical" part of this paper, or as stated in the CfP the "substantial [...] research on the theoretical aspects of Information Retrieval." I would assume that the "theory" behind the interesting phenomena shown in this case study is something like the "lexical gap" or the "vocabulary problem", concepts that are well-known to the community. But I don't see a clear connection drawn to these underlying theories. Therefore I am missing a clear discussion of the results and how these related to the underlying theories or how conclusion can be drawn.

An example for an interesting result, but a missing alignment to theory might be "Experiment 4". Seed queries perform better in comparison to the query variants. But why is this the case? Is this related to the way the original pools were constructed? Maybe a discussion around the theoretical aspects of test collection construction would strengthen the paper.

The results themselves are very interesting and UQV100 is definitely a good starting point for further work, also the "bootstraping" discussion might lead somewhere, but it's only briefly mentioned.

I therefore suggest that the paper should include (some) of the following:
- The paper is missing recent work on including (simulations of) query variants into the evaluation process (two recent ECIR papers).
- There is no discussion how neural/dense/LLM retrieval methods in relation to "classic" sparse retrieval (that is used here)
- The title is overselling: "A Concerning Case Study" might be too much here...

---

### Official Review · Reviewer_gHLk · 2024-05-15

**Rating:** -1
**Confidence:** 5

**Objective Part Of Review:**

The paper addresses questions about query variability: representing the same information need using different queries. The main finding as I see it as that topic titles (TREC) are not necessarily effective representatives of information needs. The paper is well written and easy to follow.

**Subjective Part Of Review:**

There has been much discussion of title queries versus query variants in the paper: Information Needs, Queries, and Query Performance Prediction. SIGIR 2019: 395-404. This paper is not mentioned here. Some of their findings were that title queries had median retrieval performance with respect to query variants in the UQV dataset.

There is another paper also not cited here: Topic Difficulty: Collection and Query Formulation Effects. ACM Trans. Inf. Syst. 40(1): 19:1-19:36 (2022) which studied in depth the effect of query variations on topic (information need) difficulty etc.

The authors write that: "The UQV queries pertain
to the TREC 2013 and 2014Web track, and thus provide a case study
in which the representativeness of seed queries can be tested. This
is currently the only combination of resources with which such an
experiment can be carried out". This is incorrect. Refer to the above papers for an additional collection with query variants based on ROBUST.

The findings in the paper are not surprising an echo to some extent those in previous studies not mentioned in this paper. In addition, the second "standard" dataset for query variations was not used.

---

### Official Review · Reviewer_wXqn · 2024-05-19

**Rating:** 2
**Confidence:** 5

**Objective Part Of Review:**

This is an interesting paper with implications for information retrieval research using test collections.

The authors build on the UQV data to assess whether the impact of query variability, a phenomenon already known, might affect test collections where (usually) a single query is provided per topic.

The role of the backstory remains un-investigated; these and the queries collected in response are taken as ground truth without further scrutiny. A related point then arises: the authors present their results only for using the UQV judgements, which differ from the original TREC judgement set. It would be a very useful addition to the paper if they could also report the size of the observed effects when you'd only consider the TREC judgements. I'm not suggesting these would invalidate the results, and they are likely worse (as the queries from UQV are then not developed using the same background story as the TREC assessors did) but it would still be good to have these results too.

**Subjective Part Of Review:**

The paper is well written.

I think that the authors could have written more about the relationship between previous findings in [3] and their work - readers will now have to sort that out themselves, and while I do not claim that it is an omission that harms the paper, it would improve the paper to explicitly state where the new results are complementary.

The x-axes of the figures stretch from zero to one but the interesting parts (and also referred to in the text) are at 0.01 on the x-axis, so the presentation here (say Figure 3) is imperfect when trying to follow the claims in the text.

Could you write a bit more about the effect of pooling vs the observation of experiment 5?

---

### Meta-Review · Area_Chair_2Uc3 · 2024-05-31

**Recommendation:** Reject
**Confidence:** 5

**Metareview:**

This paper uses the UQV dataset to demonstrate that TREC titles in the 2013/2014 Web Tracks are not representative queries and the result in system performance metrics and relative orderings that the other variations would disagree with.

I encourage the authors to carefully read the comments from all the reviewers as they have left thoughtful reviews with insightful comments. This paper was debated intensely and the reviewers did not reach a consensus on the recommendation.

As meta-reviewer, I'm recommending a reject for two reasons: a) the conclusion of the paper may be overstated, and b) I see significant overlap with prior work that has not been cited or addressed in this paper (see comments by reviewer gHLk). The authors should review the missing related work (especially the TOIS22 paper, "Topic Difficulty: Collection and Query Formulation Effects") and articulate the novelty of this work compared to previous publications. Details to follow:

The fact that TREC queries are not average performing queries when compared to the rest of UQV has been previously shown (see fig 1 in ICTIR19, fig 2 in SIGIR19, fig 7/8 in TOIS22) and as pointed out by reviewer cgiT, could be due to the difference in time in which those queries were constructed. Whether this is as concerning as it may first appear is unclear, given that, as some reviewers have pointed out, query rewriting and query expansion were common practices, especially when attempting to "win" TREC. :) Given this, the conclusion of the paper may be overstated.

TOIS22 also measures the impact of query formulation (and query expansions) using ANOVA and explicitly discusses the fact that query formulations have a substantial impact on relative system performances: "That is, the distribution of which topics perform well or poorly can be arbitrarily reordered using query formulations such that relative system performances change, as their performance may be better or worse depending on the specific query choice. The implications of this observation should not be underestimated." The authors should review this work and discuss the novel contributions of their paper in context of the prior art.